# Universal Attacks on Equivariant Networks

## Abstract

Adversarial attacks on neural networks perturb the input at test time in order to fool trained and deployed neural network models. Most attacks such as gradient-based Fast Gradient Sign Method (FGSM) by Goodfellow et al. (2015) and Deep-Fool by Moosavi-Dezfooli et al. (2016) are input-dependent, small, pixel-wise perturbations, and they give different attack directions for different inputs. On the other hand, universal adversarial attacks are input-agnostic and the same attack works for most inputs. Translation or rotation-equivariant neural network models provide one approach to prevent universal attacks based on simple geometric transformations. In this paper, we observe an interesting spectral property shared by all of the above input-dependent, pixel-wise adversarial attacks on translation and rotation-equivariant networks. We exploit this property to get a single universal attack direction that fools the model on most inputs. Moreover, we show how to compute this universal attack direction using principal components of the existing input-dependent attacks on a very small sample of test inputs. We complement our empirical results by a theoretical justification, using matrix concentration inequalities and spectral perturbation bounds. We also empirically observe that the top few principal adversarial attack directions are nearly orthogonal to the top few principal invariant directions.

## 1 Introduction

Neural network-based models achieve state of the art results on several speech and visual recognition tasks but these models are known to be vulnerable to various adversarial attacks. Szegedy et al. (2013) show that small, pixel-wise changes that are almost imperceptible to the human eye can make neural networks models grossly misclassify. They try to maximize the prediction error of a given model by finding a small pixel-wise perturbation using box-constrained L-BFGS. Goodfellow et al. (2015) propose the Fast Gradient Sign Method (FGSM) as a faster approach to find such an adversarial perturbation given by $x' = x + \epsilon \, \text{sign}\,(\nabla_x J(\theta, x, y))$, where $x$ denotes the input, $y$ denotes the class label, $\theta$ denotes the model parameters, and $J(\theta, x, y)$ denotes the loss function used to train the model.

In subsequent work on FGSM, its iterative variant called Projected Gradient Descent (PGD) was introduced by Kurakin et al. (2017); also see Madry et al. (2018). Goodfellow et al. (2015) and Madry et al. (2018) study adversarial perturbations from the $\ell_\infty$-ball around the input $x$, namely, each pixel value is perturbed by a quantity within $[-\epsilon, +\epsilon]$. DeepFool by Moosavi-Dezfooli et al. (2016) gives a method to compute the minimal $\ell_2$-norm perturbation. Broadly, all the above-mentioned adversarial attacks give directions to perturb any given test input in a way that is model-dependent and input-dependent. Tramer et al. (2017) also mention model-agnostic perturbations using the direction of the difference between the intra-class means, in an attempt to understand how adversarial attacks transfer across different models.

Universal adversarial attacks are input-agnostic so that the same perturbation or input transformation fools the trained model on most or nearly all test inputs. Recent work by Moosavi-Dezfooli et al. (2016; 2017) on universal adversarial attacks looks at the curvature of the decision boundary. Their universal attacks or perturbations are more sophisticated than the simple and fast, gradient-based adversarial perturbations, and require significantly more computation. Moosavi-Dezfooli et al. (2017) give a theoretical analysis of existence of universal adversarial attacks using directions in which the decision boundary has positive curvature.

Geometric transformations such as rotations, translations are simple input transformations that are model-agnostic and input-agnostic. Geometric transformations of input at test time are also possible natural attacks; see Dumont et al. (2018), Gilmer et al. (2018). One way to counter such attacks is to use neural network models that are translation and rotation-equivariant by construction. Standard Convolutional Neural Networks (StdCNNs) are translation-equivariant but not equivariant with respect to other spatial symmetries such as rotations, reflections etc. Variants of CNNs to achieve rotation-equivariance and other symmetries have received much attention recently, notably, Harmonic Networks (H-Nets) by Worrall et al. (2016), cyclic slicing and pooling by Dieleman et al. (2016), Tranformation-Invariant Pooling (TI-Pooling) by Laptev et al. (2016), Group-equivariant Convolutional Neural Networks (GCNNs) by Cohen & Welling (2016), Steerable CNNs by Cohen & Welling (2017), Deep Rotation Equivariant Networks (DREN) by Li et al. (2017), Rotation Equivariant Vector Field Networks (RotEqNet) by Marcos et al. (2017), Polar Transformer Networks (PTN) by Esteves et al. (2018). Among these, GCNNs are based on steerable filters, have a solid theoretical justification Kondor & Trivedi (2018), and achieve nearly state of the art results on MNIST-rot[1] and CIFAR10 as reported in Esteves et al. (2018). However, both StdCNNs and GCNNs do require rotation augmentation at training time to be robust to rotations.

The above discussion raises an important question: are there any interesting properties shared by translation and rotation-equivariant neural network models that can be exploited to come up with simple universal attacks? Can one find a universal attack direction using only a small fraction of test inputs but that fools the trained model on most test inputs? We answer both of these questions affirmatively in this paper.

## 2  SUMMARY OF OUR RESULTS

In this paper, we mostly study StdCNNs and GCNNs, which are translation-equivariant and rotation-equivariant, respectively. Some of our experiments also include Fully Connected NNs and RotEqNets.

- Our first observation is that even though the adversarial attack directions based on gradients, FGSM and DeepFool for StdCNN and GCNN models are input-dependent, overall they have only a small number of dominant principal components. For example, the top 5 (or $5/784 \approx 0.64\%$) principal components of the gradient directions on all test inputs contain around 10% of the total spectrum for StdCNNs, GCNNs as well as RotEqNets trained on MNIST, Fashion MNIST. For CIFAR10 the top 5 (or $5/3072 \approx 0.16\%$) principal components of the gradient directions contain $\approx 1\%$ of the total spectrum and this goes upto 3% when train and test augmented with a larger range of random rotations.

- Our second observation is that a small pixel-wise perturbation in the direction of the top principal component alone can be used as a universal attack to fool these models on around 80-90% test inputs for MNIST and Fashion MNIST, and around 60-70% test inputs for CIFAR10. And our universal attack becomes better as we train these models with larger rotations.

- Our third observation is that the top principal component can be well-approximated using only 1% sample of FGSM or DeepFool attack directions on the test data, and this approximation can still give a small pixel-wise perturbation to fool the model on around 80% of test inputs for MNIST and Fashion MNIST, and around 60% of test inputs for CIFAR10. We give a theoretical justification of this phenomenon using matrix concentration inequalities and spectral perturbation bounds. An interesting aspect of our empirical study is that these observations hold across multiple input-dependent attacks (gradient, FGSM, DeepFool) and the choice of equivariant neural network models (StdCNNs, GCNNs).

## 3  PRINCIPAL COMPONENTS OF ATTACK AND INVARIANT DIRECTIONS

We take a StdCNN or GCNN network as given in Table 2 and train it with 60,000 samples for MNIST and Fashion MNIST, and 50,000 samples for CIFAR10 training data augmented with random rotations in the range $[-180°, +180°]$. We have also performed some of the experiments for NN (fully

---

[1]http://www.iro.umontreal.ca/ lisa/twiki/bin/view.cgi/Public/MnistVariations

connected neural networks) and RotEqNet. We then use the gradient (similarly FGSM, DeepFool) directions obtained for this trained model on the 10,000 test inputs (which are also augmented with random rotations in the range $[-180°, +180°]$). We form a matrix whose rows are unit vectors along these 10,000 gradient (similarly FGSM, DeepFool) directions and obtain its Singular Value Decomposition (SVD). For gradients, we observe, as in Figure (1), that the top 5 or less than 1% in number of the squared singular values of this matrix add up to more than 10% of the overall sum of squared singular values. This is irrespective of whether the trained model is NN, StdCNN, GCNN or RotEqNet. For MNIST, Figures (18a), (2a), (3a), and (17a) show how the sigular values drop when we consider gradient, FGSM, DeepFool directions for NNs, StdCNNs, GCNNs, RotEqNets, respectively. For Fashion MNIST, Figures (21a), (19a) and (20a) show how the sigular values drop when we consider gradient, FGSM, DeepFool directions for NNs, StdCNNs, GCNNs, respectively. For CIFAR10, Figures (4a) and (5a) show how the sigular values drop when we consider gradient, FGSM, DeepFool directions for StdCNNs and GCNNs, respectively. These indicate that the drop in singular values is a common phenomenon for the different attacks and different translation and rotation-equivariant neural network models we consider.

We next consider the principal components of invariant directions. For this, the MNIST/Fashion MNIST/CIFAR10 test data is augmented with random rotations in the range $[-180°, +180°]$, and we look at the difference vectors between such images and their small $2°$ rotations. For NN, StdCNN, GCNN or RotEqNet trained with rotation augmentation, we expect the difference between a small rotation of an image and the image itself to be along the invariant subspace tangent to the level sets of the loss function. We take the principal components of these difference or *invariant* directions, and observe that the top 5 singular vectors of the adversarial directions are nearly orthogonal to the top 5 singular vectors of the invariant directions, i.e., their average dot product is roughtly less than $0.1$. Figures (18b), (2b), (3b), (17b) indicate that this property is shared by NNs, StdCNNs, GCNNs and RotEqNets for MNIST, and also holds when we look at a smaller sample of only 500 test points instead of all 10,000 test points. Similarly Figures (21b), (19b) and (20b) indicate that this property is shared by NNs, StdCNNs, GCNNs for Fashion MNIST. And Figures (4b), (5b) indicate that this property is shared by StdCNNs and GCNNs for CIFAR10. Table 1 shows deeper analysis using principal angles between subspaces that the two 5-dimensional SVD subspaces of adversarial directions and invariant directions, respectively, have nearly $90°$ principal angles.

The intensity maps of top singular vectors for the gradient, FGSM, DeepFool directions for NNs, StdCNNs, GCNNs and RotEqNets also have interesting structure as seen in Figures (36)-(54). As we have already observed, these are nearly orthogonal to the invariant directions which lead to steerable filters similar to Figure (35). We believe this underlying structure is useful and of independent interest.

## 4 UNIVERSAL ADVERSARIAL ATTACK USING A SMALL SAMPLE

We now plot the fooling rate or the fraction of test inputs that the trained model misclassifies, when we perturb all of them using the top singular vector of gradient direction scaled by $\epsilon$ that denotes the norm of perturbation. In Figures (6)-(9) we plot these fooling rate to show that we can fool 80-90% of test inputs using the same small-norm universal adversarial attack for MNIST dataset. In Figures (10)-(13) we plot these fooling rate to show that we can fool 60-70% of test inputs using

Table 1: Subspace angles between Top 5 SVD vectors from gradients of test points for each network and invariant directions (small $2°$ rotations) respectively.

| Dataset | Model | 1 | 2 | 3 | 4 | 5 |
|---|---|---|---|---|---|---|
| MNIST | StdCNN | 89.82170934 | 88.06580815 | 84.76831292 | 81.22444162 | 78.85267548 |
| MNIST | GCNN | 89.06822425 | 88.27986252 | 87.45134508 | 84.26710498 | 68.77651938 |
| MNIST | RotEqNet | 89.75561818 | 88.88988678 | 85.60567329 | 82.28177397 | 68.96832307 |
| MNIST | NN | 89.9915157 | 88.78972167 | 88.5374161 | 86.56321357 | 70.12080709 |
| Fashion MNIST | StdCNN | 89.21807572 | 79.28255649 | 77.29202168 | 70.53545269 | 63.20996031 |
| Fashion MNIST | GCNN | 88.90127175 | 87.92549838 | 73.42571367 | 70.55420976 | 69.19559671 |
| Fashion MNIST | NN | 89.9895849 | 87.71857462 | 87.5599056 | 84.38550172 | 58.77045895 |
| CIFAR10 | StdCNN | 89.97432038 | 89.91897569 | 89.84706226 | 89.71168446 | 89.56515156 |
| CIFAR10 | GCNN | 89.99666115 | 89.82916528 | 89.71339735 | 89.59987204 | 88.90306292 |

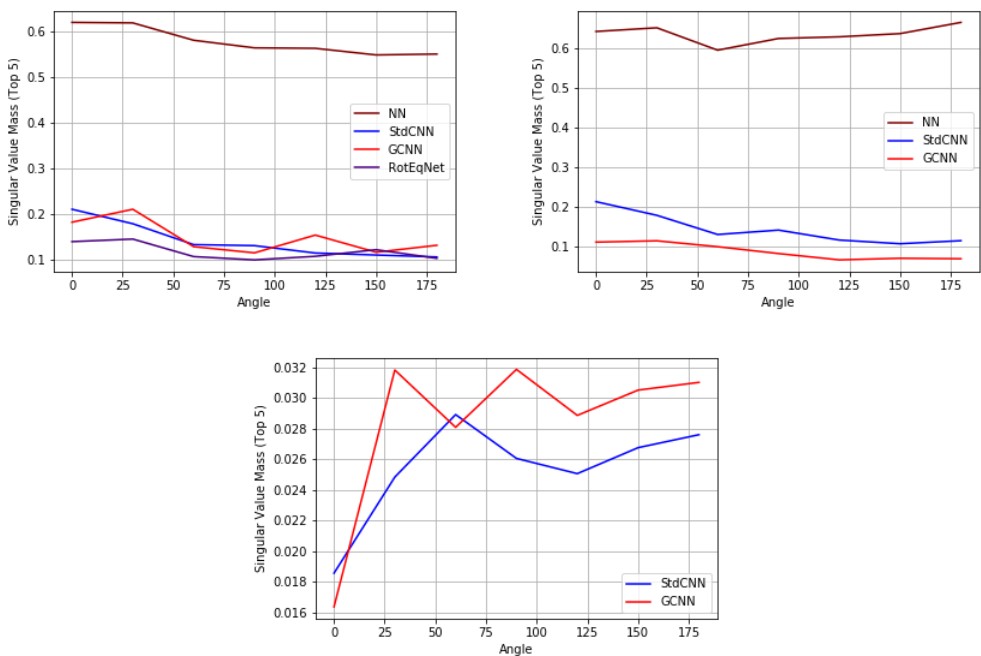

Figure 1: Fraction of the total spectrum contained in top 5 squared singular values of gradient directions for 10,000 test inputs, when train and test augmented with random rotations in $[-x°, x°]$ range for NN, StdCNN, GCNN and RotEqNet, respectively. (left) MNIST (center) Fashion MNIST (below) CIFAR10

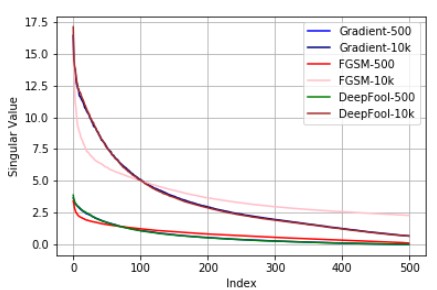

(a) Singular values of attack directions over a sample of 500 and 10,000 test points

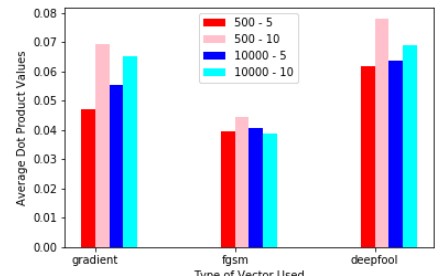

(b) Avg. dot product of top 5, top 10 singular vectors of adversarial and invariant directions, respectively, for a sample of 500 and 10,000 test points

Figure 2: On MNIST, Principal components of adversarial and invariant directions for StdCNN

the same small-norm universal adversarial attack for CIFAR10 dataset. Moreover, if we pick a small sample of 100 random test points (about 1% of test data) and take their gradients, their top singular vector also gives a good universal adversarial attack direction with a comparable fooling rate, namely, simultaneously fooling the model on roughly 80% of test points using the same small-norm perturbation for MNIST dataset and roughly 60% for CIFAR10 datset.

Please see Figures (22)-(29) in the appendix for additional experiments.

Figures (7), (9) for MNIST and (11), (13) for CIFAR10 show that a simple approach using the top singular vector of the gradient directions on a small sample gives a universal adversarial attack comparable to Moosavi-Dezfooli et al. (2017) which we denote in the plots as M-DFFF. For applying

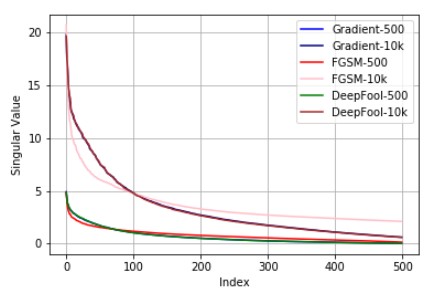
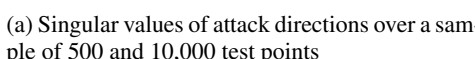

(a) Singular values of attack directions over a sample of 500 and 10,000 test points

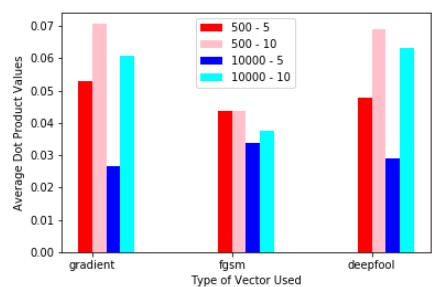

(b) Avg. dot product of top 5, top 10 singular vectors of adversarial and invariant directions, respectively, for a sample of 500 and 10,000 test points

Figure 3: On MNIST, Principal components of adversarial and invariant directions for GCNN

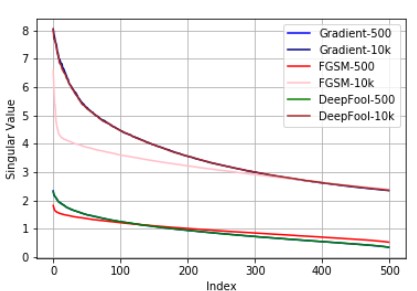

(a) Singular values of attack directions over a sample of 500 and 10,000 test points

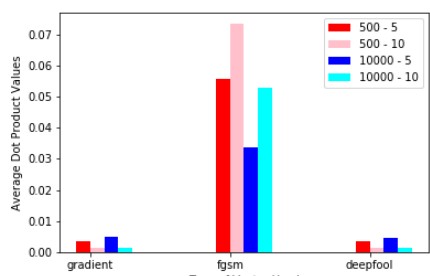

(b) Avg. dot product of top 5, top 10 singular vectors of adversarial and invariant directions, respectively, for a sample of 500 and 10,000 test points

Figure 4: On CIFAR10, Principal components of adversarial and invariant directions for StdCNN

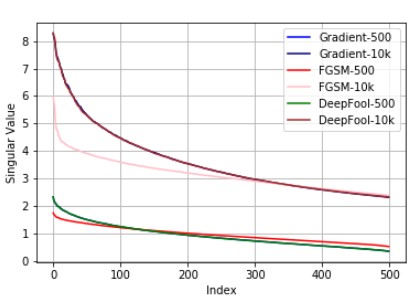

(a) Singular values of attack directions over a sample of 500 and 10,000 test points

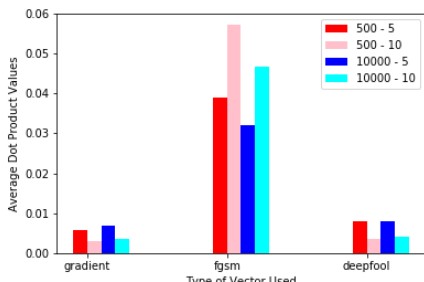

(b) Avg. dot product of top 5, top 10 singular vectors of adversarial and invariant directions, respectively, for a sample of 500 and 10,000 test points

Figure 5: On CIFAR10, Principal components of adversarial and invariant directions for GCNN

M-DFFF we take the perturbation got by their method, scale it to a unit vector and apply to the test set with varying norm of perturbation to plot the fooling rate.

Figures (14)-(16) and (30)-(34) show how the fooling rate of our adversarial attack gets better when the training is augmented with larger rotations.

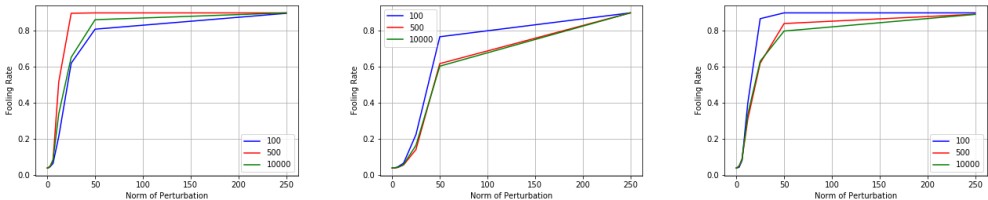

Figure 6: On MNIST, StdCNN: fooling rate vs. norm of perturbation along top singular vector of attack directions on 100/500/10000 sample, (left) gradients (center) FGSM (right) DeepFool

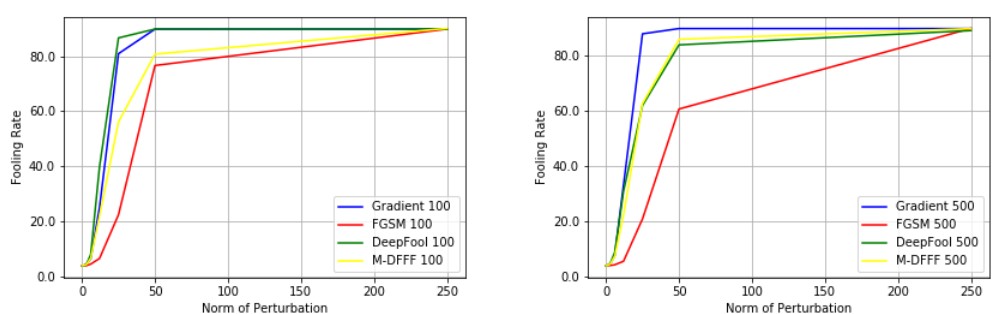

Figure 7: On MNIST, StdCNN: fooling rate vs. norm of perturbation along top singular vector of attack directions on, (left) 100 samples, (right) 500 samples

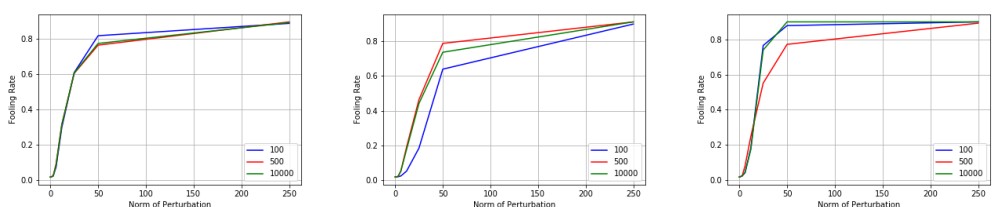

Figure 8: On MNIST, GCNN: fooling rate vs. norm of perturbation along top singular vector of attack directions on 100/500/10000 samples, (left) gradients (center) FGSM (right) DeepFool

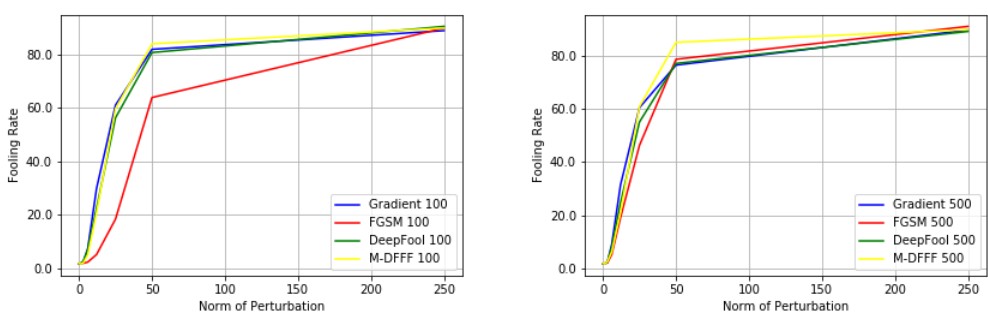

Figure 9: On MNIST, GCNN: fooling rate vs. norm of perturbation along top singular vector of attack directions on, (left) 100 samples, (right) 500 samples

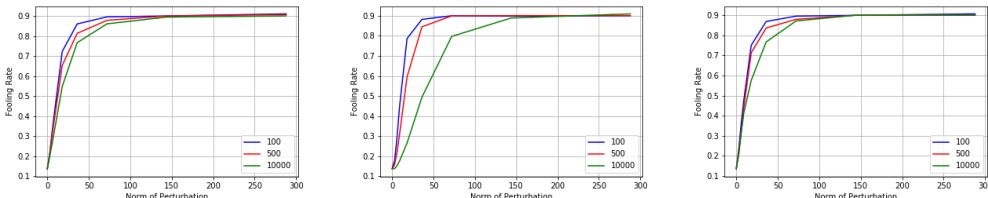

Figure 10: On CIFAR10, StdCNN: fooling rate vs. norm of perturbation along top singular vector of attack directions on 100/500/10000 sample, (left) gradients (center) FGSM (right) DeepFool

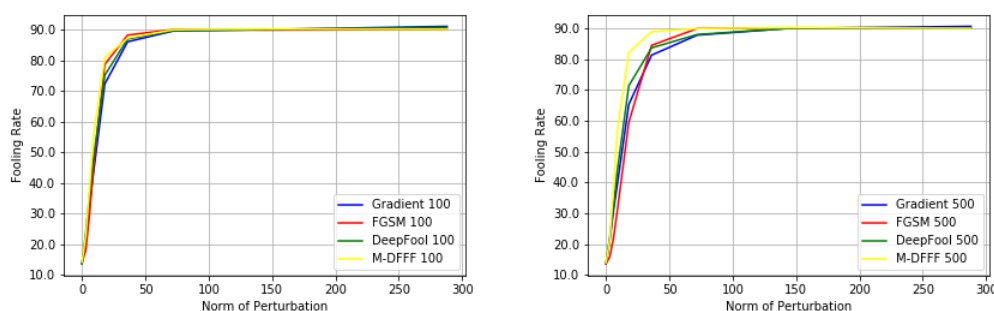

Figure 11: On CIFAR10, StdCNN: fooling rate vs. norm of perturbation along top singular vector of attack directions on, (left) 100 samples, (right) 500 samples

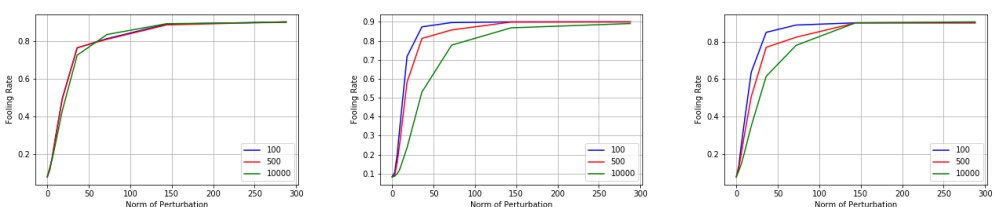

Figure 12: On CIFAR10, GCNN: fooling rate vs. norm of perturbation along top singular vector of attack directions on 100/500/10000 sample, (left) gradients (center) FGSM (right) DeepFool

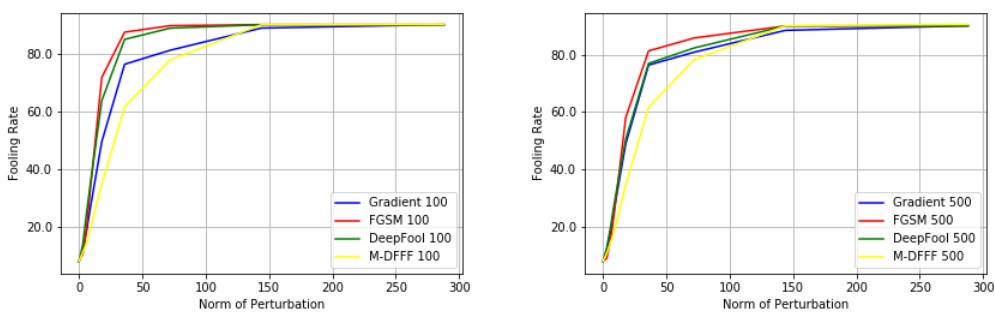

Figure 13: On CIFAR10, GCNN: fooling rate vs. norm of perturbation along top singular vector of attack directions on, (left) 100 samples, (right) 500 samples

## 5 ANALYSIS OF UNIVERSAL ADVERSARIAL PERTURBATIONS

In this section, we attempt to provide a theoretical justification for the existence of universal adversarial perturbations. Assume that our data is in $d$ dimensions $X \subseteq \mathbb{R}^d$ with an underlying

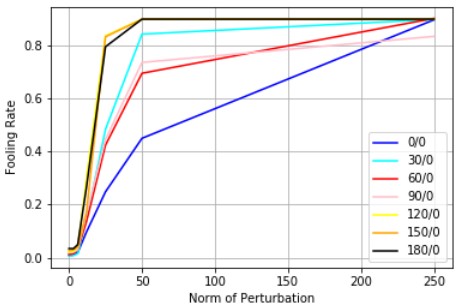 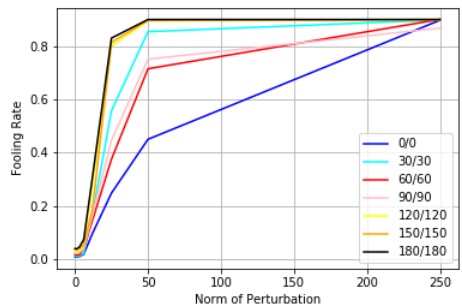

Figure 14: On MNIST, fooling rate of our universal attack using top singular vector of 100 test point gradients, for StdCNN train-augmented with random rotations in range $[-\theta°, \theta°]$ (left) test unrotated, (right) test-augmented with the same range of rotations as the training set.

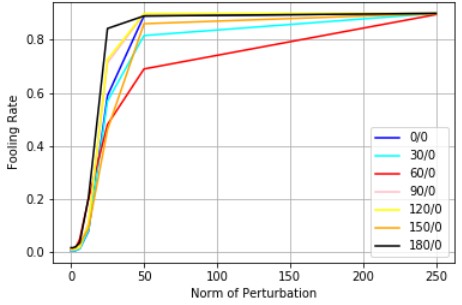 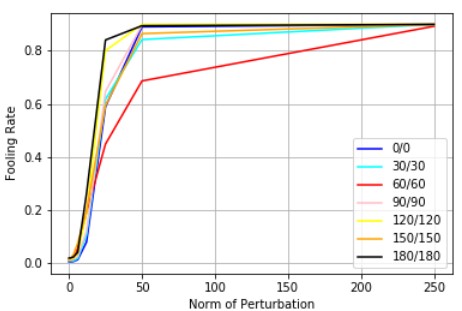

Figure 15: On MNIST, fooling rate of our universal attack using top singular vector of 100 test point gradients, for GCNN train-augmented with random rotations in range $[-\theta°, \theta°]$ (left) test unrotated, (right) test-augmented with the same range of rotations as the training set.

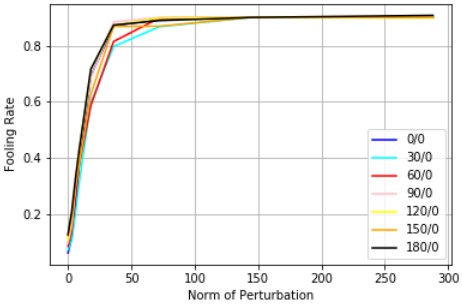 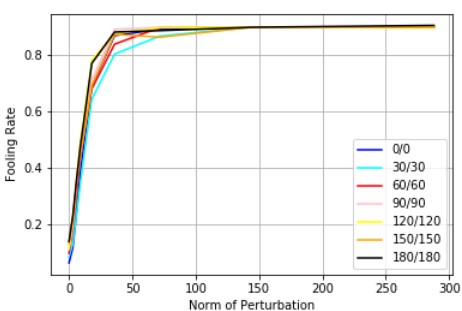

Figure 16: On CIFAR10, fooling rate of our universal attack using top singular vector of 100 test point gradients, for StdCNN train-augmented with random rotations in range $[-\theta°, \theta°]$ (left) test unrotated, (right) test-augmented with the same range of rotations as the training set.

distribution given by probability density $\mu(\mathbf{x})$ at each point $\mathbf{x} \in X$. Let $\mathbf{a_x}$ be an adversarial perturbation direction for each $\mathbf{x}$, respectively. Note that $\mathbf{a_x}$ could be obtained by any of the several available methods such as the Fast Gradient Sign Method (FGSM), the Kurakin attack, DeepFool or by finding the nearest point on the decision boundary etc. Two important questions are: (a) Why does there exist a *universal* vector or direction that works as an adversarial perturbation for many or

most data points simultaneously? (b) Can we compute such a *universal* adversarial perturbation by looking only at a small subsample of the data?

Given any method that obtains adversarial perturbations for individual data points, if the matrix of these adversarial perturbation directions taken over all data points satisfies a certain property, then we show that the top singular vectors of this matrix are good candidates for *universal* adversarial perturbations that make many points to be misclassified simultaneously.

**Theorem 1.** *Let $X \subseteq \mathbb{R}^d$ be any given input data with the underlying probability density $\mu(\mathbf{x})$ at $\mathbf{x} \in X$. Let $\mathbf{a_x}$ denote an adversarial perturbation vector at point $\mathbf{x} \in X$. Let $0 \leq \lambda \leq 1$ be the top eigenvalue of the matrix $M \in \mathbb{R}^{d \times d}$ defined as*

$$M = \mathsf{E}\left[\frac{\mathbf{a_x}}{\|\mathbf{a_x}\|_2}\frac{\mathbf{a_x}^T}{\|\mathbf{a_x}\|_2}\right],$$

*and let $\mathbf{v}$ be its corresponding eigenvector. Then*

$$\mathsf{Pr}\left(\{\mathbf{x} \; : \; |\langle \mathbf{a_x}, \mathbf{v}\rangle| \geq \delta \; \|\mathbf{a_x}\|_2\}\right) \geq \frac{\lambda - \delta^2}{1 - \delta^2}.$$

*In particular, plugging in $\delta = \sqrt{\lambda/2}$ we get*

$$\mathsf{Pr}\left(\left\{\mathbf{x} \; : \; |\langle \mathbf{a_x}, \mathbf{v}\rangle| \geq \sqrt{\frac{\lambda}{2}} \; \|\mathbf{a_x}\|_2\right\}\right) \geq \frac{\lambda}{2}.$$

*Proof.* Define $S = \{\mathbf{x} \; : \; |\langle \mathbf{a_x}, \mathbf{v}\rangle| \geq \delta \; \|\mathbf{a_x}\|_2\} \subseteq X$. Since $\lambda$ is the top eigenvalue of $M$ with $\mathbf{v}$ as its corresponding eigenvector,

$$
\begin{aligned}
\lambda &= \int_X \left\langle \frac{\mathbf{a_x}}{\|\mathbf{a_x}\|_2}, \mathbf{v}\right\rangle^2 \mu(\mathbf{x})d\mathbf{x} \\
&= \int_{\mathbf{x} \in S} \left\langle \frac{\mathbf{a_x}}{\|\mathbf{a_x}\|_2}, \mathbf{v}\right\rangle^2 \mu(\mathbf{x})d\mathbf{x} + \int_{\mathbf{x} \notin S} \left\langle \frac{\mathbf{a_x}}{\|\mathbf{a_x}\|_2}, \mathbf{v}\right\rangle^2 \mu(\mathbf{x})d\mathbf{x} \\
&\leq \int_{\mathbf{x} \in S} \mu(\mathbf{x})d\mathbf{x} + \delta^2 \int_{\mathbf{x} \notin S} \mu(\mathbf{x})d\mathbf{x} \qquad\qquad \text{because } \|\mathbf{v}\|_2 = 1 \\
&= \mathsf{Pr}\,(S) + \delta^2\,(1 - \mathsf{Pr}\,(S)) \\
&= (1 - \delta^2)\,\mathsf{Pr}\,(S) + \delta^2.
\end{aligned}
$$

This implies that $\mathsf{Pr}\,(S) \geq (\lambda - \delta^2)/(1 - \delta^2)$. $\qquad\square$

Observe that $\mathrm{tr}\,(M) = 1$. Theorem 1 implies that as the top eigenvalue $\lambda$ dominates the spectrum, its eigenvector $\mathbf{v}$ get more aligned with the adversarial perturbations directions $\mathbf{a_x}/\|\mathbf{a_x}\|_2$ for most points $\mathbf{x} \in X$. This means that $\mathbf{v}$ is a potential candidate for a *universal* adversarial perturbation. More generally, Theorem 1 works for any of the top eigenvalues of $M$ and their corresponding eigenvectors, and thus, gives a subspace spanned by multiple orthogonal directions all of which are potential candidates for being *universal* adversarial perturbations that fool a given classifier on many input examples.

Now let's consider the second question of finding a good approximation to our candidate *universal* adversarial perturbation or the top eigenvector $\mathbf{v}$ of $M$ using only a small subsample of $X$. Theorem 2 shows that only a small sample of size independent of $|X|$ but depending only on the dimension $d$ of the data and the spectral properties of the matrix $M$ suffices.

**Theorem 2.** *Let $X \subseteq \mathbb{R}^d$ be any given input data with the underlying probability density $\mu(\mathbf{x})$ at $\mathbf{x} \in X$. Let $\mathbf{a_x}$ denote an adversarial perturbation vector at point $\mathbf{x} \in X$. Let $0 \leq \lambda \leq 1$ be the top eigenvalue of the matrix $M \in \mathbb{R}^{d \times d}$ defined as*

$$M = \mathsf{E}\left[\frac{\mathbf{a_x}}{\|\mathbf{a_x}\|_2}\frac{\mathbf{a_x}^T}{\|\mathbf{a_x}\|_2}\right],$$

*and let $\mathbf{v}$ be its corresponding eigenvector. Let $\mathbf{x}_1, \mathbf{x}_2, \ldots, \mathbf{x}_m$ be a i.i.d. sample of $m$ points from $X$ drawn from the distribution defined by probability density $\mu$, and let $\tilde{\lambda}$ be the top eigenvalue of the matrix $\tilde{M}$,*

$$\tilde{M} = \frac{1}{m} \sum_{i=1}^{m} \frac{\mathbf{a}_{\mathbf{x}_i}}{\|\mathbf{a}_{\mathbf{x}_i}\|_2} \frac{\mathbf{a}_{\mathbf{x}_i}^T}{\|\mathbf{a}_{\mathbf{x}_i}\|_2},$$

*and $\tilde{\mathbf{v}}$ be the top eigenvector of $\tilde{M}$.*

*Also suppose that there is a gap of at least $\gamma\lambda$ between the top eigenvalue $\lambda$ and the second eigenvalue of $M$. Then for any $0 \le \epsilon < \gamma$ and $m = O(\epsilon^{-2} d \log d)$, we get $\|\mathbf{v} - \tilde{\mathbf{v}}\|_2 \le \epsilon/(\gamma - \epsilon)$, with a constant probability. This probability can be boosted to $1 - \delta$ by having an additional $\log(1/\delta)$ in the $O(\cdot)$.*

*Proof.* Using matrix Bernstein inequality from Tropp (2015), we get that for $m = O(\epsilon^{-2} d \log d)$, we have that the spectral or the operator norm $\left\| M - \tilde{M} \right\|_2 \le \epsilon\lambda$, with a constant probability. Now we can use results about spectral perturbation bounds from Bhatia (1997). By Weyl's theorem, this implies $\left| \lambda - \tilde{\lambda} \right| \le \epsilon\lambda$. If there is gap of at least $\gamma\lambda$ between the first and the second eigenvalue of $M$ with $\gamma > \epsilon$, then by Davis-Kahan theorem we get that $\|\mathbf{v} - \tilde{\mathbf{v}}\|_2 \le \epsilon/(\gamma - \epsilon)$, with a constant probability. $\qquad\square$

## 6 CONCLUSION

We show how to use a small sample of input-dependent adversarial attack directions on test inputs to find a single universal attack direction that fools translation and rotation-equivariant models such as CNNs and GCNNs on a large fraction of test inputs. Our main observation is a spectral property shared by different attacks directions such as gradients, FGSM, DeepFool on these models. We give a theoretical justification for how this spectral property helps in using the top singular vector as a universal attack direction computed using only a small sample of test inputs.

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

## A   DETAILS OF EXPERIMENTS

All experiments performed on neural network-based models were done using MNIST, Fashion MNIST and CIFAR10 datasets with appropriate augmentations applied to the train/validation/test set.

**Data sets**   MNIST[2] dataset consists of $70,000$ images of $28 \times 28$ size, divided into 10 classes. $55,000$ used for training, $5,000$ for validation and $10,000$ for testing. Fashion MNIST[3] dataset consists of $70,000$ images of $28 \times 28$ size, divided into 10 classes. $55,000$ used for training, $5,000$ for validation and $10,000$ for testing. CIFAR10[4] dataset consists of $60,000$ images of $32 \times 32$ size, divided into 10 classes. $40,000$ used for training, $10,000$ for validation and $10,000$ for testing.

---

[2]http://www.iro.umontreal.ca/ lisa/twiki/bin/view.cgi/Public/MnistVariations
[3]https://github.com/zalandoresearch/fashion-mnist
[4]https://www.cs.toronto.edu/ kriz/cifar.html

| Standard CNN | GCNN |
|---|---|
| Conv(10,3,3) + Relu | P4ConvZ2(10,3,3) + Relu |
| Conv(10,3,3) + Relu | P4ConvP4(10,3,3) + Relu |
| Max Pooling(2,2) | Group Spatial Max Pooling(2,2) |
| Conv(20,3,3) + Relu | P4ConvP4(20,3,3) + Relu |
| Conv(20,3,3) + Relu | P4ConvP4(20,3,3) + Relu |
| Max Pooling(2,2) | Group Spatial Max Pooling(2,2) |
| FC(50) + Relu | FC(50) + Relu |
| Dropout(0.5) | Dropout(0.5) |
| FC(10) + Softmax | FC(10) + Softmax |

Table 2: Architectures used for experiments

**Model Architectures**   For the MNIST and Fashion MNIST based experiments we use the 7 layer architecture of GCNN similar to Cohen & Welling (2016). The StdCNN architecture is similar to the GCNN except that the operations are as per CNNs. Refer to Table 2 for details. RotEqNet architecture is as given in Marcos et al. (2017). The NN architecture is a 2 layer fully connected with 784-50 neurons in the layers with dropout. For the CIFAR10 based experiments we use the ResNet18 architecture as in He et al. (2016) and it's equivalent in GCNN as given in Cohen & Welling (2016). Input training data was augmented with random cropping and random horizontal flips, apart from the specific range of rotation augmentation as needed.

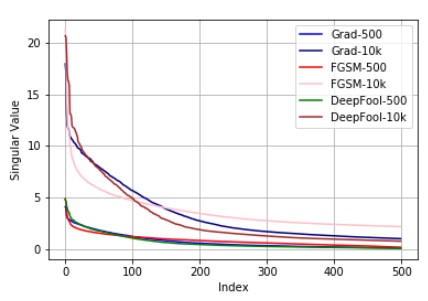

(a) Singular values of attack directions over a sample of 500 and 10,000 test points

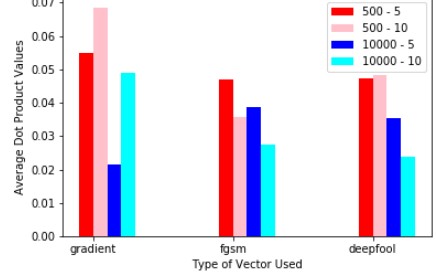

(b) Avg. dot product of top 5, top 10 singular vectors of adversarial and invariant directions, respectively, for a sample of 500 and 10,000 test points

Figure 17: On MNIST, Principal components of adversarial and invariant directions for RotEqNet

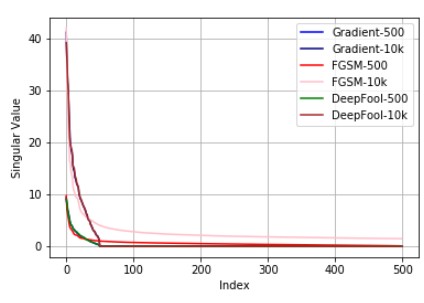

(a) Singular values of attack directions over a sample of 500 and 10,000 test points

(b) Avg. dot product of top 5, top 10 singular vectors of adversarial and invariant directions, respectively, for a sample of 500 and 10,000 test points

Figure 18: On MNIST, Principal components of adversarial and invariant directions for NN

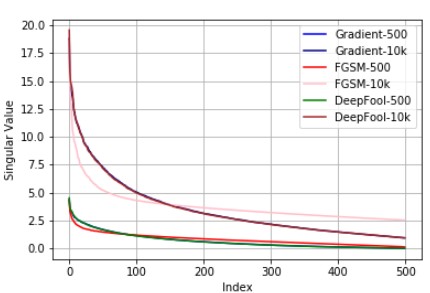 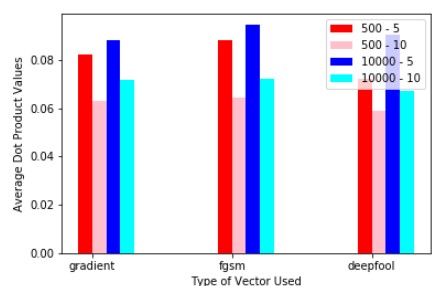

(a) Singular values of attack directions over a sample of 500 and 10,000 test points

(b) Avg. dot product of top 5, top 10 singular vectors of adversarial and invariant directions, respectively, for a sample of 500 and 10,000 test points

Figure 19: On Fashion MNIST, Principal components of adversarial and invariant directions for StdCNN

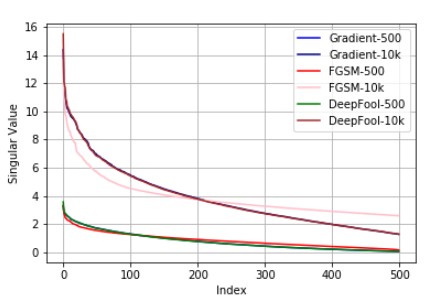 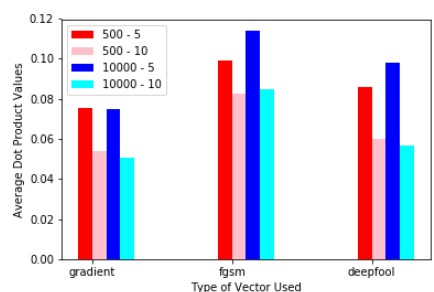

(a) Singular values of attack directions over a sample of 500 and 10,000 test points

(b) Avg. dot product of top 5, top 10 singular vectors of adversarial and invariant directions, respectively, for a sample of 500 and 10,000 test points

Figure 20: On Fashion MNIST, Principal components of adversarial and invariant directions for GCNN

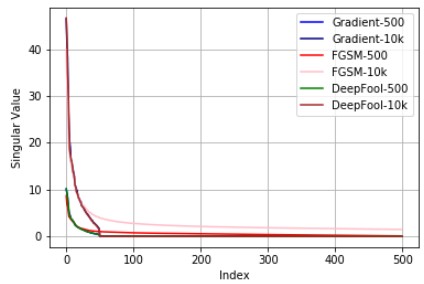

(a) Singular values of attack directions over a sample of 500 and 10,000 test points

(b) Avg. dot product of top 5, top 10 singular vectors of adversarial and invariant directions, respectively, for a sample of 500 and 10,000 test points

Figure 21: On Fashion MNIST, Principal components of adversarial and invariant directions for NN

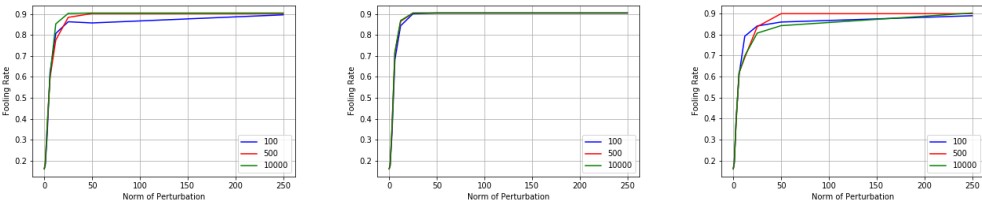

Figure 22: On MNIST, NN: fooling rate vs. norm of perturbation along top singular vector of attack directions on 100/500/10000 sample, (left) gradients (center) FGSM (right) DeepFool

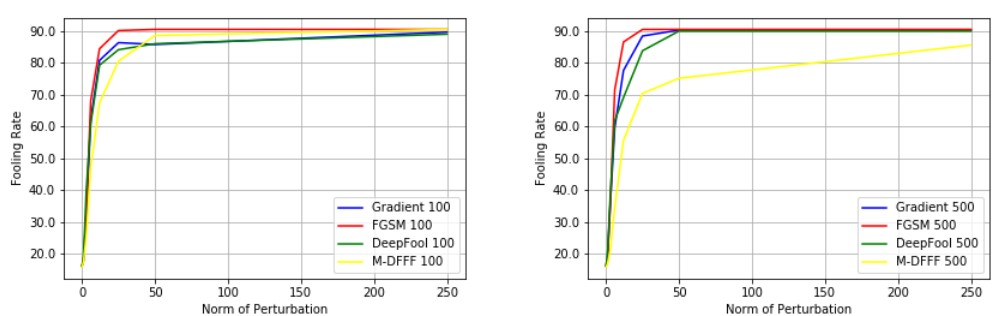

Figure 23: On MNIST, NN: fooling rate vs. norm of perturbation along top singular vector of attack directions on, (left) 100 samples, (right) 500 samples

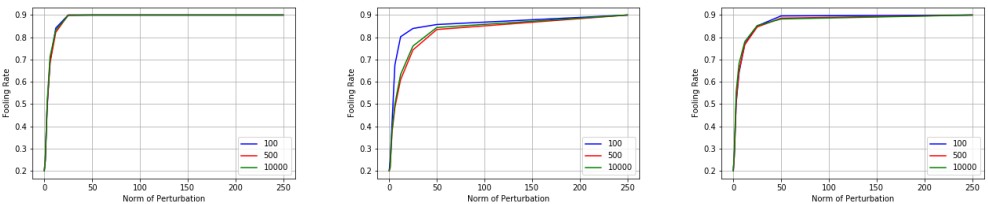

Figure 24: On Fashion MNIST, StdCNN: fooling rate vs. norm of perturbation along top singular vector of attack directions on 100/500/10000 sample, (left) gradients (center) FGSM (right) Deep-Fool

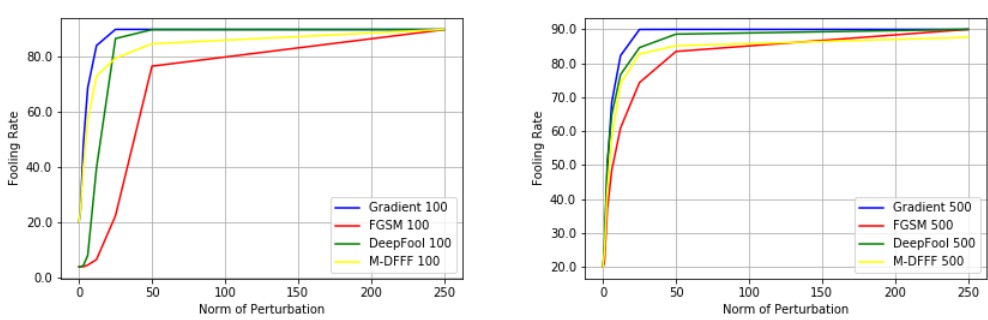

Figure 25: On Fashion MNIST, StdCNN: fooling rate vs. norm of perturbation along top singular vector of attack directions on, (left) 100 samples, (right) 500 samples

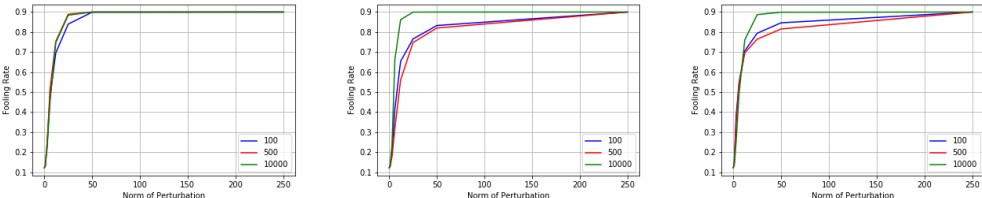

Figure 26: On Fashion MNIST, GCNN: fooling rate vs. norm of perturbation along top singular vector of attack directions on 100/500/10000 sample, (left) gradients (center) FGSM (right) Deep-Fool

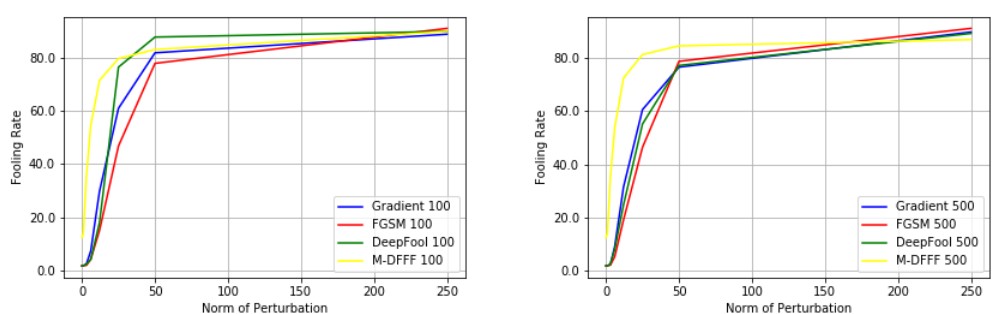

Figure 27: On Fashion MNIST, GCNN: fooling rate vs. norm of perturbation along top singular vector of attack directions on, (left) 100 samples, (right) 500 samples

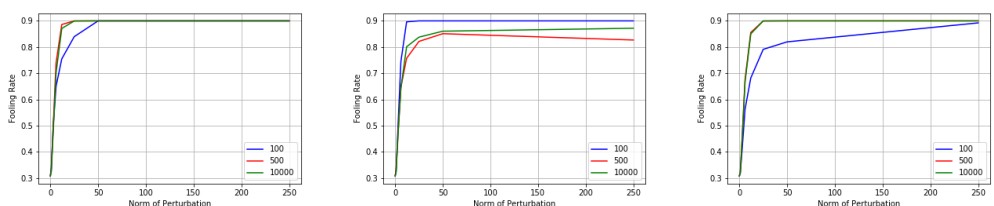

Figure 28: On Fashion MNIST, NN: fooling rate vs. norm of perturbation along top singular vector of attack directions on 100/500/10000 sample, (left) gradients (center) FGSM (right) DeepFool

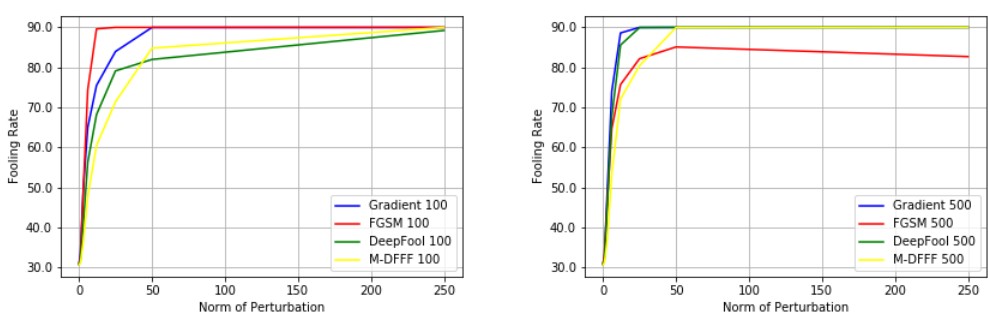

Figure 29: On Fashion MNIST, NN: fooling rate vs. norm of perturbation along top singular vector of attack directions on, (left) 100 samples, (right) 500 samples

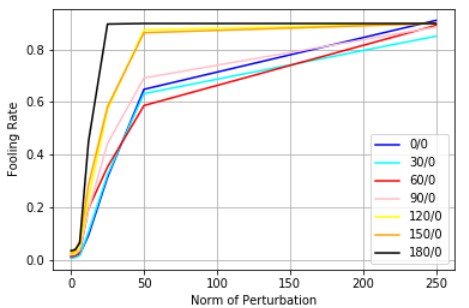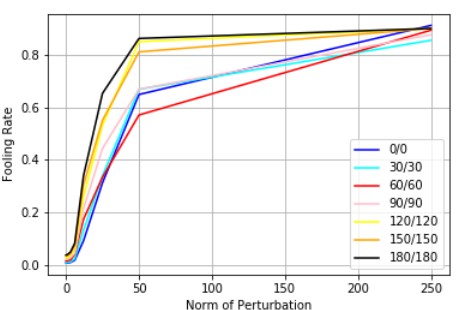

Figure 30: On MNIST, fooling rate of our universal attack using top singular vector of 10000 test point gradients, for StdCNN train-augmented with random rotations in range $[-\theta°, \theta°]$ (left) test unrotated, (right) test-augmented with the same range of rotations as the training set.

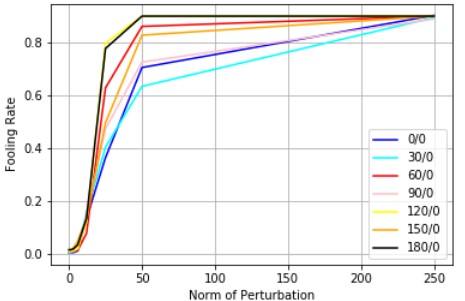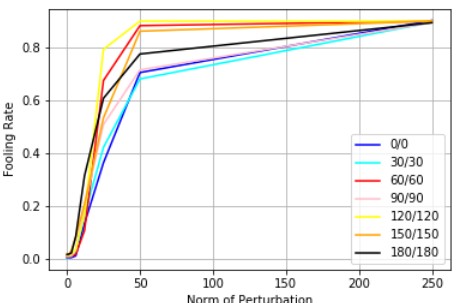

Figure 31: On MNIST, fooling rate of our universal attack using top singular vector of 10000 test point gradients, for GCNN train-augmented with random rotations in range $[-\theta°, \theta°]$ (left) test unrotated, (right) test-augmented with the same range of rotations as the training set.

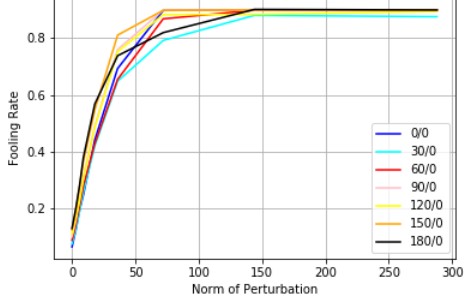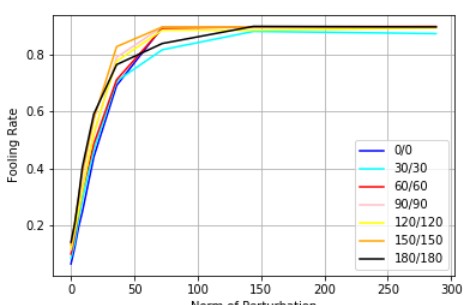

Figure 32: On CIFAR10, fooling rate of our universal attack using top singular vector of 10000 test point gradients, for StdCNN train-augmented with random rotations in range $[-\theta°, \theta°]$ (left) test unrotated, (right) test-augmented with the same range of rotations as the training set.

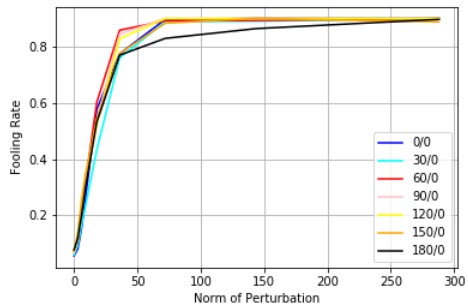 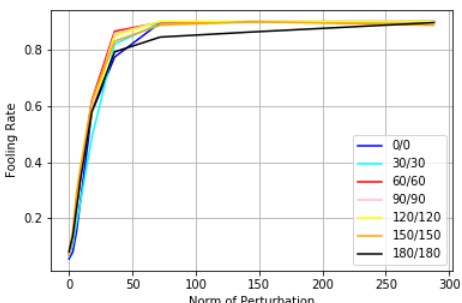

Figure 33: On CIFAR10, fooling rate of our universal attack using top singular vector of 100 test point gradients, for GCNN train-augmented with random rotations in range $[-\theta°, \theta°]$ (left) test unrotated, (right) test-augmented with the same range of rotations as the training set.

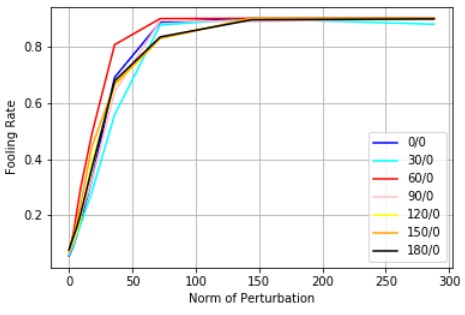 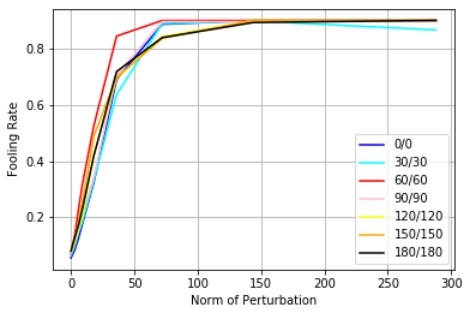

Figure 34: On CIFAR10, fooling rate of our universal attack using top singular vector of 10000 test point gradients, for GCNN train-augmented with random rotations in range $[-\theta°, \theta°]$ (left) test unrotated, (right) test-augmented with the same range of rotations as the training set.

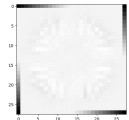 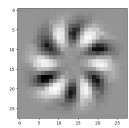 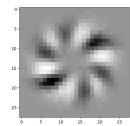 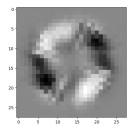 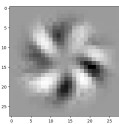

Figure 35: On MNIST, Top 5 SVD vectors from image difference

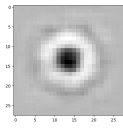 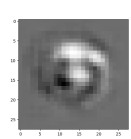 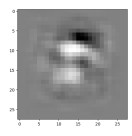 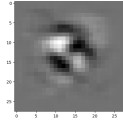 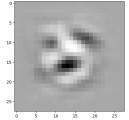

Figure 36: On MNIST, Top 5 SVD vectors from Gradients in StdCNN with rotation augmentations

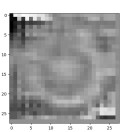 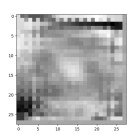 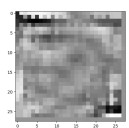 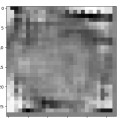 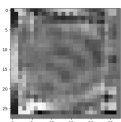

Figure 37: On MNIST, Top 5 SVD vectors from FGSM in StdCNN with rotation augmentations

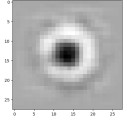 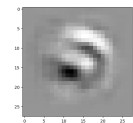 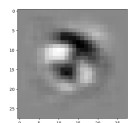 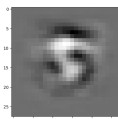 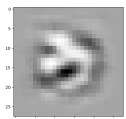

Figure 38: On MNIST, Top 5 SVD vectors from DeepFool in StdCNN with rotation augmentations

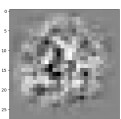 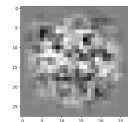

Figure 39: On MNIST, M-DFFF perturbation vectors for StdCNN with rotation augmentations, with (left) 100 samples (right) 500 samples

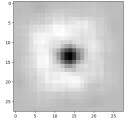 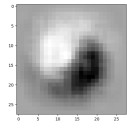 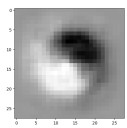 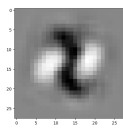 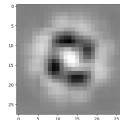

Figure 40: On MNIST, Top 5 SVD vectors from Gradients in GCNN with rotation augmentations

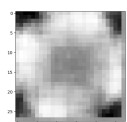 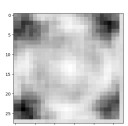 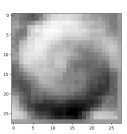 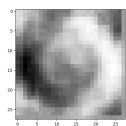 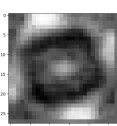

Figure 41: On MNIST, Top 5 SVD vectors from FGSM in GCNN with rotation augmentations

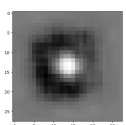 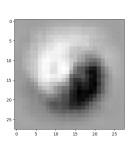 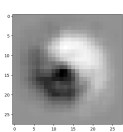 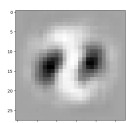 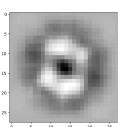

Figure 42: On MNIST, Top 5 SVD vectors from DeepFool in GCNN with rotation augmentations

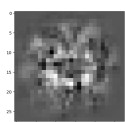 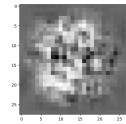

Figure 43: On MNIST, M-DFFF perturbation vectors for GCNN with rotation augmentations, with (left) 100 samples (right) 500 samples

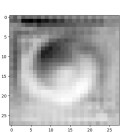 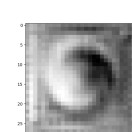 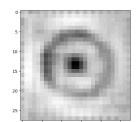 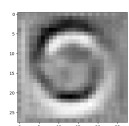 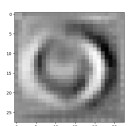

Figure 44: On MNIST, Top 5 SVD vectors from Gradients in RotEqNet with rotation augmentations

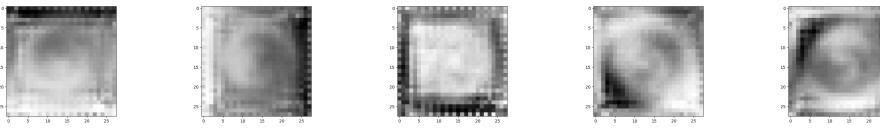

Figure 45: On MNIST, Top 5 SVD vectors from FGSM in RotEqNet with rotation augmentations

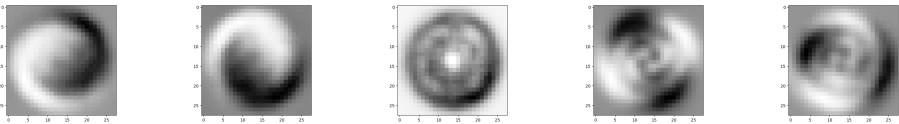

Figure 46: On MNIST, Top 5 SVD vectors from DeepFool in RotEqNet with rotation augmentations



Figure 47: On MNIST, Top 5 SVD vectors from Gradients in NN with rotation augmentations

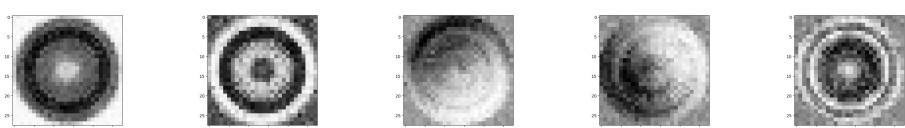

Figure 48: On MNIST, Top 5 SVD vectors from FGSM in NN with rotation augmentations



Figure 49: On MNIST, Top 5 SVD vectors from DeepFool in NN with rotation augmentations

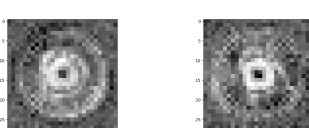

Figure 50: On MNIST, M-DFFF perturbation vectors for NN with rotation augmentations, with (left) 100 samples (right) 500 samples

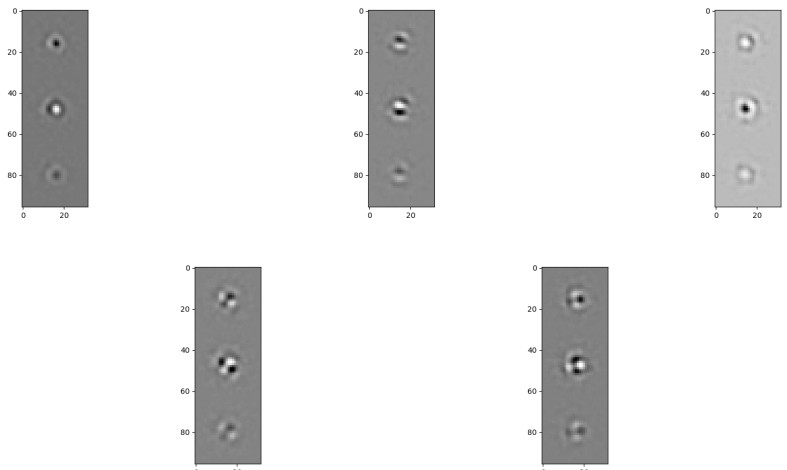

Figure 51: On CIFAR10, Top 5 SVD vectors from Gradients in StdCNN with rotation augmentations

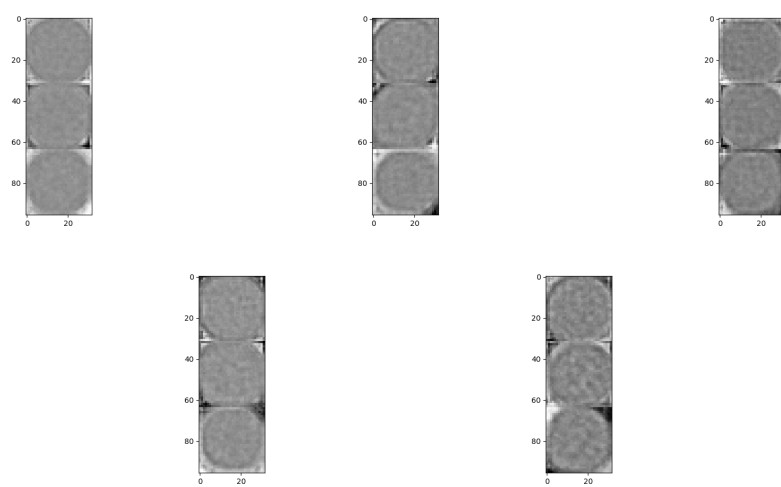

Figure 52: On CIFAR10, Top 5 SVD vectors from FGSM in StdCNN with rotation augmentations

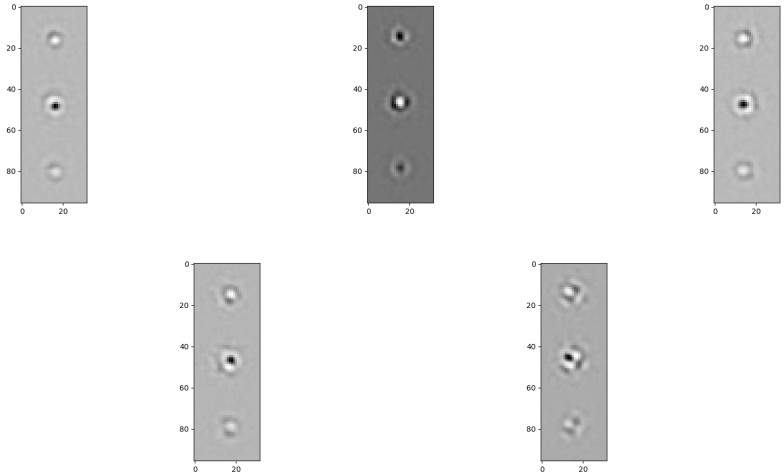

Figure 53: On CIFAR10, Top 5 SVD vectors from DeepFool in StdCNN with rotation augmentations



Figure 54: On CIFAR10, M-DFFF perturbation vectors for StdCNN with rotation augmentations, with (left) 100 samples (right) 500 samples