# OpenReview forum: "Universal Attacks on Equivariant Networks"
_ICLR.cc/2019/Conference_

### Official Review · AnonReviewer3 · 2018-10-31
**Interesting observations, but paper needs clarity in writing**

**Rating:** 5
**Confidence:** 4

**Review:**

The paper presents some interesting observations related to the connection between the universal adversarial attacks on CNNs and spectral properties. While most of the results are empirical, the authors present two theorems to justify some of the observations. However, the paper is poorly written and very hard to read. Rather than providing too many plots/results in the main paper (maybe use supplementary matl.), the empirical results should be better explained to help the readers. Similarly, the implications of the theorems are not really clear and bit hand-wavy.

xxxxxxxxxxxxxx

It seems that the authors provided a generic response to all the reviewers and I am not sure if they acknowledge the lack of clarity and lot of hand-wavy explanations in the paper. This issue has been raised by other reviewers too and is quite critical for becoming a good paper worthy for ICLR. Therefore, I am unable to update my score for this paper. However, I do appreciate the comparison with Moosavi-Dezfooli et al. (CVPR'17), this is a good addition as suggested by another reviewer.

---

> ### Author Response · Authors · 2018-11-26
> **Revised with CIFAR10 experiments and comparison with Moosavi-Dezfooli et al.**
>
> We show that the principal attack directions are nearly orthogonal to the principal invariant directions. Models learns invariance to rotations either when we explicitly use an equivariant network (GCNN, RotEqNet) or when we train any model (StdCNN, fully connected NN) with rotation augmentations or do both. We show a simple universal adversarial attack using the top principal component of any input-dependent attack direction on a small test sample. We show that even a simple approach of using the top singular vector of the gradients on a small sample of test points is comparable to the attack of Moosavi-Dezfooli et al. (CVPR'17). Moreover, the fooling rate of our universal attack gets better as the model is train-augmented with larger rotations.

---

### Official Review · AnonReviewer1 · 2018-11-04
**An interesting observation, but the contribution is not significant enough**

**Rating:** 4
**Confidence:** 5

**Review:**

The authors made an interesting observation: There's an important common subspace of Gradient/FGSM/Deepfool attacks among all examples. Therefore, they propose to use top SVD components of the directions to conduct universal attack. This is an interesting finding but also not surprising; we know the gradient of loss function w.r.t input can be used for interpretability, and in MNIST examples they usually reveals some rough shape of the class. This is also observed in Figure 8-13 in this paper, and thus it makes sense that the gradient directions share a common subspace. Therefore I think this observation itself is not significant enough.

Using this for universal attack is interesting, however the experiments are not that convincing:

1. To show this is a good way for universal attack, I think the authors should compare with previous work in (Moosavi-Dezfooli et al).

2. All the experiments are on MNIST. How about cifar/ImageNet?

---

> ### Author Response · Authors · 2018-11-26
> **Revised with CIFAR10 experiments and comparison with Moosavi-Dezfooli et al.**
>
> We have modified the submission with the experiments on CIFAR 10 dataset asked by the reviewer.
>
> We show that the principal attack directions are nearly orthogonal to the principal invariant directions. Models learns invariance to rotations either when we explicitly use an equivariant network (GCNN, RotEqNet) or when we train any model (StdCNN, fully connected NN) with rotation augmentations or do both. We show a simple universal adversarial attack using the top principal component of any input-dependent attack direction on a small test sample. We show that even a simple approach of using the top singular vector of the gradients on a small sample of test points is comparable to the attack of Moosavi-Dezfooli et al. (CVPR'17). Moreover, the fooling rate of our universal attack gets better as the model is train-augmented with larger rotations.

---

### Official Review · AnonReviewer2 · 2018-11-05
**Principal directions towards universal attacks**

**Rating:** 4
**Confidence:** 5

**Review:**

This paper studies the problem of computing non-data specific perturbations, also known as universal perturbations, to attack neural networks and take profit of their inherent vulnerability. Compared to previous works in the domain, the authors look specifically at equivariant networks, and derive geometric insights and methods to compute universal perturbations for these networks.

The paper starts by analysing the main/principal directions of set of perturbations that are able to change the decisions in different forms of equivariant neural networks. With this heuristic study, a few main directions are shown to be shared by most adversarial perturbations. The authors then propose to construct universal perturbations built on the insights given by the principle directions of perturbations, which is an interesting an effective method. In addition, it is shown that a few adversarial samples are sufficient to identify pretty  accurately the principle directions. The fooling rates achieved by this method is pretty good, which demonstrates that the proposed strategy is reasonable.

The key idea in this paper (using principal shared directions of perturbations, computed on a small subset of data points) has unfortunately already been proposed and tested in classical (non-equivariant) neural networks - see for example Fig 9 in Moosavi-Dezfooli, 2017, cited in the paper, and published in CVPR 2017. The present paper proposes however a few additional bits of information with a nice theoretical analysis, while the previous works were mostly based on heuristics. It is probably not sufficient however to pass the cut in ICLR.

The interesting additional novelty here is the study of equivariant networks. However, this ends up falling sort of initial expectations - there seems to be nothing specific to equivariant networks in the proposed study, and the solution and algorithm is actually applicable to any neural network architectures (?). Also, no specific insights are derived for equivariant networks, which could be potentially very interesting to make progress in understanding better equivariant representations, which still consist in a widely open research problem.

In general, the paper has a non-classical organisation, with a lot of heuristics that are not discussed in depth - that gives a sort of high-level impression that the proposed idea is potentially nice, but that but superficially addressed. It should probably be improved in the next versions of this work.

---

> ### Author Response · Authors · 2018-11-26
> **Revised with CIFAR10 experiments and comparison with Moosavi-Dezfooli et al.**
>
> We show that the principal attack directions are nearly orthogonal to the principal invariant directions. Models learns invariance to rotations either when we explicitly use an equivariant network (GCNN, RotEqNet) or when we train any model (StdCNN, fully connected NN) with rotation augmentations or do both. We show a simple universal adversarial attack using the top principal component of any input-dependent attack direction on a small test sample. We show that even a simple approach of using the top singular vector of the gradients on a small sample of test points is comparable to the attack of Moosavi-Dezfooli et al. (CVPR'17). Moreover, the fooling rate of our universal attack gets better as the model is train-augmented with larger rotations.

---

### Meta-Review · Area_Chair1 · 2018-12-15
**Some interesting contribution, but there are significant exposition (and novelty) concerns**

**Confidence:** 4
**Recommendation:** Reject

**Metareview:**

The topic of universal adversarial perturbation is quite intriguing and fairly poorly studied and the paper provides a mix of new insights, both theoretical and empirical in nature. However, the significant presentation issues make it hard to properly understand and evaluate them. In particular, the theoretical part feels rushed and not sufficiently rigorous, and it is unclear why focusing on the case of equivariant network is crucial. Also, it would be useful if the authors put more effort in explaining how their contributions fit into the context of prior work in the area.

Overall, this paper has a potential of becoming a solid contribution, once the above shortcomings are addressed.